# Metastatic Neuroendocrine Neoplasms of Unknown Primary: Clues from Pathology Workup

**DOI:** 10.3390/cancers14092210

**Published:** 2022-04-28

**Authors:** Carl Christofer Juhlin, Jan Zedenius, Anders Höög

**Affiliations:** 1Department of Oncology-Pathology, Karolinska Institutet, 171 64 Stockholm, Sweden; anders.hoog@ki.se; 2Department of Pathology and Cancer Diagnostics, Karolinska University Hospital, 171 76 Stockholm, Sweden; 3Department of Molecular Medicine and Surgery, Karolinska Institutet, 171 64 Stockholm, Sweden; jan.zedenius@ki.se; 4Department of Breast, Endocrine Tumors and Sarcoma, Karolinska University Hospital, 171 76 Stockholm, Sweden

**Keywords:** neuroendocrine neoplasm, neuroendocrine tumor, metastatic, unknown primary, morphology, immunohistochemistry

## Abstract

**Simple Summary:**

While most neuroendocrine neoplasms are indolent and slow-growing tumors, subsets of cases will spread beyond the tissue of origin. Given the rather slow progress, some lesions are incidentally discovered as metastatic deposits rather than primary masses. In these cases, a biopsy is often taken to allow the pathologist to identify the tumor type and possibly the primary tumor site via microscopic examination. In this review, the authors present a simplified guide on how to approach metastatic neuroendocrine tumors from a pathologist’s perspective.

**Abstract:**

Neuroendocrine neoplasms (NENs) are diverse tumors arising in various anatomical locations and may therefore cause a variety of symptoms leading to their discovery. However, there are instances in which a NEN first presents clinically as a metastatic deposit, while the associated primary tumor is not easily identified using conventional imaging techniques because of small primary tumor sizes. In this setting (which is referred to as a “NEN of unknown primary”; NEN-UP), a tissue biopsy is often procured to allow the surgical pathologist to diagnose the metastatic lesion. If indeed a metastatic NEN-UP is found, several clues can be obtained from morphological assessment and immunohistochemical staining patterns that individually or in concert may help identify the primary tumor site. Herein, histological and auxiliary analyses of value in this context are discussed in order to aid the pathologist when encountering these lesions in clinical practice.

## 1. Introduction

Although most primary tumors in cases of neuroendocrine neoplasia (NEN) are discovered in the clinical workup of patients with symptoms either directly or indirectly associated with the tumor burden, the rather slow growth rate of these lesions allows for incidental discoveries of a metastatic mass with a lack of a primary site mass upon imaging analyses [1,2,3]. So-called NENs of unknown primary (NEN-UPs) are not entirely uncommon in the clinical setting and may constitute 12–22% of NEN patients [4,5,6]. However, as the treatment and prognosis of NENs may vary with the tissue of origin, it is imperative to allocate the primary tumor in order to modify the clinical handling [6]. Indeed, the primary site is a significant factor in terms of overall prognosis when consulting large NEN patient datasets, which may also be reflected by the fact that patients with NEN-UPs have the poorest overall survival of all NEN patients [4,7]. If the primary site is not determined with certainty, therapeutic strategies are usually based on tumor grading, hormonal activity/functionality, somatostatin receptor (SSTR) status, as well as the tumor burden [6].

NENs represent approximately 0.5–1% of all tumors, and the incidence is estimated to be between approximately 2–4/100,000, with a female preponderance [8,9]. In terms of tumor grade, NEN-UPs are most often well-differentiated grade 1 or 2 tumors [7] and most commonly originate from the intestinal system (approximately 60–65% of cases) or lungs (approximately 20–25%) [6,8]. A computerized tomography (CT) scan has a rather high sensitivity to detect a primary NEN, while magnetic resonance imaging (MRI) is considered a gold standard to characterize hepatic metastases [10,11]. However, an MRI is not as helpful as a CT scan to aid in the detection of small intestinal NENs but is adequate in visualizing pancreatic NENs [12]. Specific imaging such as Gallium-68-somatostatin receptor-positron emission tomography (PET)/CT also often identifies the primary tumor whenever a NEN-UP is diagnosed, as in liver metastasis, but not always [2,13]. Additionally, PET/CT using other tracers may sometimes be of value in this setting [14]. Despite these advances in imaging techniques, there may be instances when analyses coupled with the core-needle biopsy from the metastatic lesion are required for the identification of the primary tumor, and this review targets foremost the practicing pathologists in order to provide a summarized guide on how to address the issue of metastatic NENs of unknown primary, especially concerning how the diagnostic workup could be optimized. 

### 1.1. The Core-Needle Biopsy: Advantages to Fine-Needle Aspiration Cytology

Once a metastatic deposit has been identified and the decision taken to procure biopsy material, there are several options for the clinical team to consider. Liver metastases dominate in the clinical setting, and these lesions are usually reachable using either a fine-needle aspiration biopsy (FNAB) or a core-needle biopsy (CNB), and only occasionally are focal liver resections needed to obtain sufficient material. However, as current guidelines require histological assessment of the tumor tissue in order to properly grade a NEN, a core-needle biopsy is strongly recommended [6,15]. Indeed, even though the value of FNAB is not to be entirely neglected [16,17,18], the literature supports the use of CNB rather than FNAB when assessing NENs (both primary and metastatic) [19,20,21,22]. In CNB material, the Ki-67 index can usually be established by counting at least 2000 cells in hotspot areas, and other advantages include the possibility to perform immunohistochemistry to pinpoint the tumoral origin, a possible hormone production, the SSTR status, and potentially the PD-L1 status before considering eventual adjuvant treatment. Additionally, the advent of molecular testing has also led to a need for extensive biopsy material, and an important challenge for the pathologist is to prioritize the tissue efficiently. 

### 1.2. Diagnostic Workup and Tumor Grading: Morphology Is Key

Although modern pathology practice almost always involves immunohistochemical and molecular analyses to aid in the assessment of NENs, there are still important morphological aspects that should not be overlooked when assessing metastatic cases. This is especially true when grading the tumor, as the overall histology may be the deciding factor when assessing highly proliferative NENs that may either be considered NET G3 or, alternatively, NEC [23]. Moreover, NENs may exhibit different histological patterns also depending on the site of origin, and these parameters could therefore be important clues in terms of identifying the primary tumor when NEN-UPs arise.

NEN grading was previously recommended to only follow cutoffs for the Ki-67 labeling and mitotic indices, whereas this has changed with recent updates in the World Health Organization (WHO) classification of NENs [23]. Nowadays, morphology is also an important factor when assessing whether a lesion should qualify as a grade 3 neuroendocrine tumor (G3 NET) or NEC, in which G3 NETs display well-differentiated histology while NECs often display poor differentiation [23]. Therefore, a modern pathologist must also be trained in morphological assessment of various NENs, and not solely rely on auxiliary markers for correct grading. 

In terms of the site of origin, NENs may exhibit variable growth patterns and cellular characteristics easily identifiable on routine hematoxylin–eosin staining alone. For example, while metastatic NENs primary in the stomach and duodenum may demonstrate a glandular-like pattern, small intestinal NENs (SI-NENs) often exhibit an organoid growth pattern. In contrast, pancreatic and rectal NENs may present with a ribbon-like, trabecular architecture [24,25]. Moreover, cellular attributes may also vary, as metastatic pheochromocytomas and abdominal paragangliomas (PPGLs) present with a basophilic and intensely granular cytoplasm, while other NENs may be more amphophilic in nature [26]. Even though morphological attributes may vary between a primary tumor and its subsequent metastatic deposits, the growth pattern and cellular characteristics are still important parameters that should be considered not only in terms of designating a tumor as neuroendocrine per se but also for considering the tissue of origin when assessing NEN-UPs. 

There might also be additional site-specific hints waiting to be discovered in hematoxylin–eosin-stained preparations; for example, stromal ossification may suggest a lung primary, although this phenomenon has also been reported in NENs of extrapulmonary origin [27]. Similarly, amyloid deposits should raise the suspicion of medullary thyroid carcinoma (MTC), and the amyloid could be verified using a Congo Red stain [28]. However, it should be mentioned that other NENs also may present with amyloid, so the proper identification of a metastatic MTC will require immunohistochemical verification. In addition, psammoma bodies may occasionally be noted in somatostatinomas [29]. Some classical histological phenotypes for various NEN categories are detailed in Table 1.

### 1.3. Diagnostic Workup: Immunohistochemistry

Although histomorphology can indicate a NEN, the diagnosis usually requires immunohistochemistry to rule out potential mimics [30]. The workup should therefore include classic markers such as Chromogranin A (CGA) and Synaptophysin (SYP). While traditionally considered a marker of neuroendocrine differentiation, CD56 should be avoided in this context, given its nonspecific attributes [31,32]. Moreover, caution must be taken when assessing CGA and SYP stains, as (1) subsets of poorly differentiated NECs may stain negative for one or both markers as part of the dedifferentiation process, and (2) non-NEN tumors may occasionally stain partly or diffusely positive for these markers [33]. Therefore, the addition of second-generation neuroendocrine markers ISL1, INSM1, and Secretagogin (SECG) has been proven useful to aid in the diagnostic workup, especially when CGA or SYP stains are equivocal [34]. In broad terms, SYP is considered highly sensitive for a NEN origin, which is not least reflected in the ability of high-grade NECs to retain SYP while not seldomly losing CGA expression [30]. On the other hand, CGA is considered highly specific, as numerous non-NEN types may exhibit focal or widespread SYP immunoreactivity (adrenocortical tumors, malignant melanoma, and sarcoma) [30,35]. Cytokeratin expression should also be investigated in all NENs, as a cytokeratin-negative lesion staining positive for neuroendocrine markers may indicate a pheochromocytoma or paraganglioma [36]. 

### 1.4. Pinpointing the Primary Origin: Clues from Immunohistochemistry

In order to identify the true origin of a NEN-UP, a wide variety of immunohistochemical markers may be assessed. To facilitate the process, authors have proposed simplified schemes for clinical purposes built on the site-specific expression of various proteins. The key markers for each entity are listed in Table 2.

#### 1.4.1. Metastatic Pancreatic Neuroendocrine Neoplasia (Pan-NEN)

In terms of NEN-UPs arising in the upper gastrointestinal (GI) system, a combination of classic neuroendocrine markers, PDX1 and CDX2, may be useful to identify these lesions (Figure 1) [37]. Moreover, specific stainings for islet hormones (insulin, glucagon, somatostatin) and gastrin may also facilitate the identification of pancreatic and duodenal NENs (Figure 1). PDX1 and CDX2 are two transcription factors involved in the regulation of pancreatic islet hormone gene activation and genes expressed in the intestinal epithelium, respectively. PAX8, an additional transcription factor, may also stain positive in pancreatic NEN [38].

#### 1.4.2. Metastatic Neuroendocrine Neoplasia from Small Intestine and Appendix

NENs arising in the small intestine (SI-NEN) and appendix are usually serotonin-producing and therefore identifiable using serotonin immunohistochemistry. Moreover, most NENs derived from these anatomical sites stain diffusely for INSM1, SECG, CDX2, and SATB2, while being consistently negative for ISL1 (Figure 2) [33,39,40,41,42].

#### 1.4.3. Metastatic Colorectal NEN

Colorectal NENs often express classical neuroendocrine markers (although rectal NENs recurrently display absent CGA staining). In addition, these lesions may present with positive staining for glucagon-like peptide 1, peptide YY, CDX2, SATB2, and occasionally also PAX8 (Figure 3) [39,43,44,45].

#### 1.4.4. Metastatic Pulmonary NEN

NEN-UPs derived from the bronchi or lungs are usually positive for classic neuroendocrine markers and often express TTF1 and bombesin/gastrin-releasing peptide (GRP) [30]. Subsets of cases may also express calcitonin or calcitonin-related peptide [46]. However, note that subsets of typical and atypical carcinoids may display aberrant expression of serotonin, while pulmonary NECs may upregulate PAX8 (Figure 4) [47,48]. A recent algorithm suggested that NENs positive for TTF1 and CK7 while displaying negativity for SSR2, CDX2, and nuclear beta-catenin most often have their origin in the pulmonary system, while the opposite immune phenotype may indicate a NEN arising in the gastrointestinal system [39].

#### 1.4.5. Metastatic Pheochromocytoma and Abdominal Paraganglioma

Pheochromocytomas and abdominal paragangliomas (collectively abbreviated as PPGLs) stain positive for neuroendocrine markers CGA, SYP, ISL1, and INSM1, but not for secretagogin [26,33]. Moreover, these lesions are almost always keratin negative and often GATA3 positive, and subsets of cases may display an intricate network of supporting sustentacular cells, which are highlighted by an S100 or SOX10 stain (Figure 5) [26,36,49]. Subsets of PPGLs associated with mutations in genes regulating pseudo-hypoxic pathways may stain aberrantly positive for CAIX or alpha-inhibin. As the latter marker is also a marker of adrenal cortical differentiation, alpha-inhibin should not be used in the context of differentiating these two entities [26,50].

#### 1.4.6. Metastatic Merkel Cell Carcinoma

Merkel cell carcinomas (MCCs) are neuroendocrine skin lesions famous for their small round blue cell morphology and paranuclear dot-like CK20 positivity, but a positive Merkel cell polyomavirus stain may also help in the identification (Figure 6) [51,52,53]. Although MCCs may occasionally present as a NEN-UP, there is also the risk of confusing primary MCC with a cutaneous NEN metastasis from a nonskin origin [54]. Additionally, subsets of MCCs may arise in adjunction to a mucosal lining such as the oral cavity, and clinical workup may therefore be negative in rare instances if only dermatological investigations are pursued [55].

#### 1.4.7. A Word of Warning 

Despite all the guidelines and recommendations mentioned above, it is crucial to recognize that many transcription factors used for primary tissue identification may either up- or downregulate their expression in poorly differentiated NECs, which could trick the pathologist from a diagnostic perspective [40]. For example, TTF1 is known to show positive staining in various NECs unrelated to the bronchi/lung and thyroid [56]. It is worth mentioning that PAX8 may stain differently in various tumor types depending on monoclonal or polyclonal antibodies being used [57]. Therefore, one should not put too much emphasis on a single marker in the context of a poorly differentiated NEC but rather apply a careful approach when estimating the tissue of origin in these instances. This is also true for subsets of well-differentiated NETs, as tumors primary to the lower GI may display positive TTF1 expression while being CDX2 negative. Therefore, a combined assessment using clues from clinical history, radiology, morphology, and immunohistochemistry is advised when assessing NEN-Ups of any kind, rather than blind trust in a single marker.

### 1.5. Pinpointing the Primary Origin: Clues from Molecular Analyses

In modern medicine, auxiliary molecular testing is gaining ground as a complementary analysis to aid in therapeutic decision making, as mutational screening panels may identify actionable variants in NENs not responsive to conventional treatments [23,58]. However, there are also diagnostic benefits of utilizing next-generation sequencing (NGS) in clinical routine, as NENs developing in different tissues may have disparate genetic backgrounds and thereby be of help when assessing NEN-UPs [59]. This is mostly true for poorly differentiated NECs that may show absent staining for conventional neuroendocrine markers, as well as display aberrant expressional patterns of transcription factors, thereby possibly confusing the pathologist. Moreover, interrogating whether an NEC exhibits actionable mutations or not is also gaining ground as a complimentary clinical analysis whenever patients progress through the standard treatment.

NECs of various origins often display *TP53* or *RB1* gene aberrations, which in turn may be visualized using immunohistochemistry for these markers (Figure 7) [59,60]. Indeed, mutations in any of these genes could favor an NEC diagnosis in cases with equivocal histology and borderline Ki-67 labeling indices. NETs of various sites usually exhibit a more diverse palette of mutational signatures. Pulmonary NETs usually harbor mutations in *MEN1*, *PIK3CA*, *ARID1A,* or *KRAS*, while pancreatic NETs often display *MEN1*, *DAXX*, *ZFHX3,* or *ATRX* aberrances [23,61,62]. While pulmonary and pancreatic NENs seem to be overrepresented in mutational events in histone-modifying and chromatin-remodeling genes, small intestinal NETs are more cell cycle and Wnt pathway-driven lesions, not seldomly exhibiting *CDKN1B*, *MEN1*, *CTNNB1*, or *APC* mutations [63,64,65]. Given these differences in molecular background, it is therefore expected that the ongoing implementation of NGS in routine clinical practice will provide the endocrine pathologist with an additional toolbox for more efficient identification of tumoral origin.

## 2. Discussion

Pinpointing a neuroendocrine phenotype in a metastatic tumor is imperative in order to obtain the correct treatment, and the development of neuroendocrine markers of the first and second generation has greatly facilitated this procedure [30,34]. However, careful histological and immunohistochemical investigations may also help to identify the tissue of origin in cases in which the patients present with a metastatic deposit only [30]. There are morphological, immunohistochemical, and molecular clues that the practicing pathologist needs to be aware of in order to increase the likelihood of identifying the location of a primary tumor, some of the most crucial of which are reviewed herein. Although the interpretation of the markers requires a critical appraisal of the staining outcome in relation to the clinical history and the overall morphology, there are still a few cornerstones in diagnostic pathology that we believe are of direct value when interpreting NEN-Ups: (1) identify an epithelial origin to exclude pheochromocytoma/paraganglioma; (2) screen widely with TTF1, PDX1 and CDX2; and (3) verify with site-specific stainings (Figure 8). If applied stringently, most NEN-UPs would probably be assigned a correct primary location using this crude approach, although there are numerous pitfalls described for the abovementioned markers.

## 3. Conclusions

Although imaging analyses often identify the primary location of a NEN-UP, there are instances in which the primary tumor remains undetermined. In such instances, the surgical pathologist plays an important role in assessing morphology and immunohistochemical profiles, which may help identify the true origin of these lesions, which in turn may affect treatment options for the individual patient. However, care must be taken not to rely solely on the significance of a single marker, especially since the aberrant expression is common in poorly differentiated NECs.

## Figures and Tables

**Figure 1 cancers-14-02210-f001:**
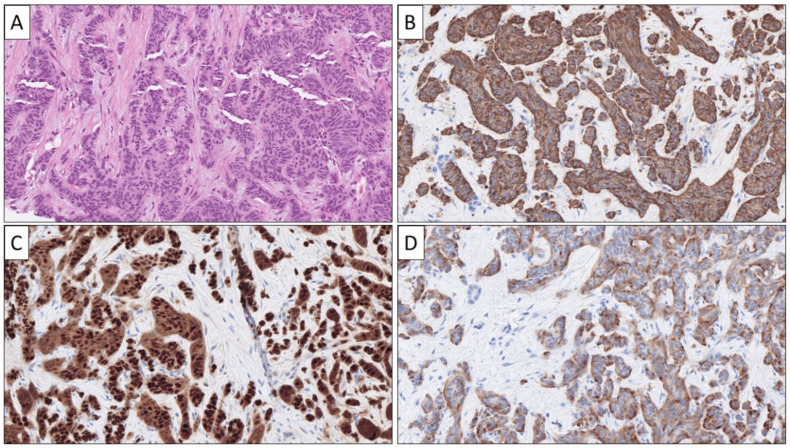
The morphological and expressional phenotype of a hormone-producing pancreatic neuroendocrine tumor (Pan-NET) metastatic to the liver. This tumor was sampled via a core-needle biopsy and exhibited a cord-like and trabecular growth pattern on routine staining (**A**) and was positive for Chromogranin A (**B**), PDX1 (**C**), and somatostatin (**D**). Take note that pancreatic and duodenal somatostatinomas may exhibit identical expressional profiles, and clinical correlation is usually required to differentiate when presenting as a NEN-UP.

**Figure 2 cancers-14-02210-f002:**
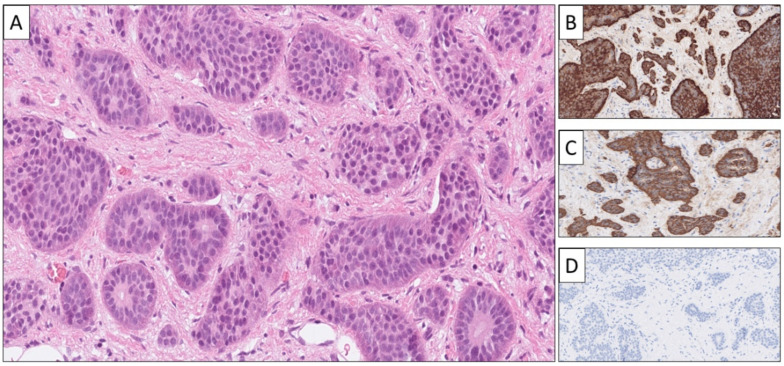
Histological and immunohistochemical attributes of a metastatic small intestinal NET (SI-NET). This lesion was core-needle biopsied from the liver, and the tumor displayed an organoid growth pattern against a fibrotic stroma on routine hematoxylin–eosin stain (**A**), with eosinophilic, cytoplasmic granulations clearly visible. Immunohistochemical expression was noted for Chromogranin A (**B**) and serotonin (**C**), while ISL1 was negative (**D**).

**Figure 3 cancers-14-02210-f003:**
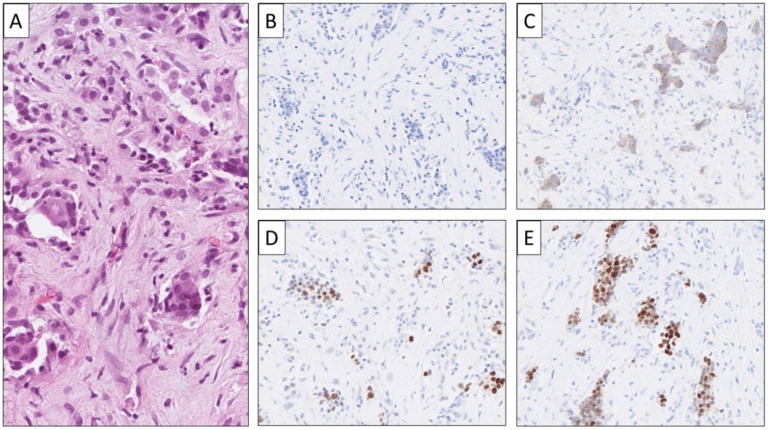
NEN-UP metastatic to the liver subsequently identified as rectal neuroendocrine carcinoma. Via a core-needle biopsy of the liver, the histological assessment was consistent with a high-grade lesion displaying solid to loose tumor cell aggregates with pleomorphic features and abundant mitotic figures (**A**). The tumor was negative for Chromogranin A (**B**) but positive for Synapthophysin (**C**). Subsets of tumor cells expressed CDX2 (**D**) and were positive for SATB2 (**E**), indicating a lower GI tract origin (colorectum).

**Figure 4 cancers-14-02210-f004:**
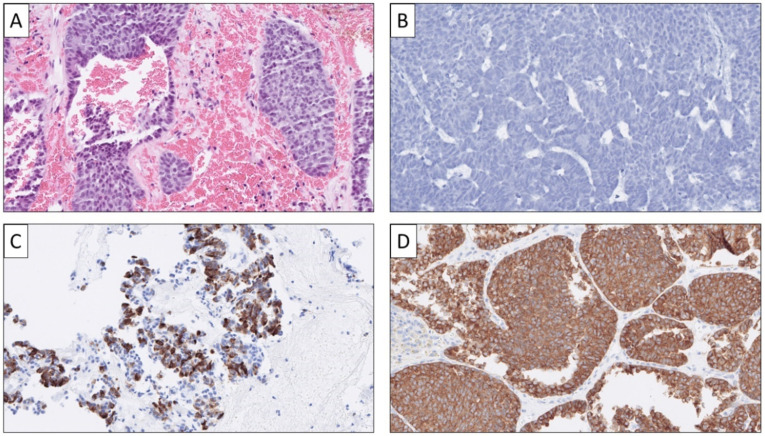
Pitfalls in the diagnostic workup of metastatic lung NENs. This lesion presented as a NEN-UP metastatic to the liver. A core-needle biopsy was performed, and histological examination revealed a nested tumor with little nuclear atypia (**A**). The tumor cells are clearly negative for TTF1 (**B**). However, serotonin was focally expressed, initially raising the suspicion of a metastatic SI-NET (**C**). Neuroendocrine markers (as exemplified by Synaptophysin in (**D**) were diffusely positive. A clinical workup identified a primary atypical lung carcinoid, which also expressed aberrant serotonin.

**Figure 5 cancers-14-02210-f005:**
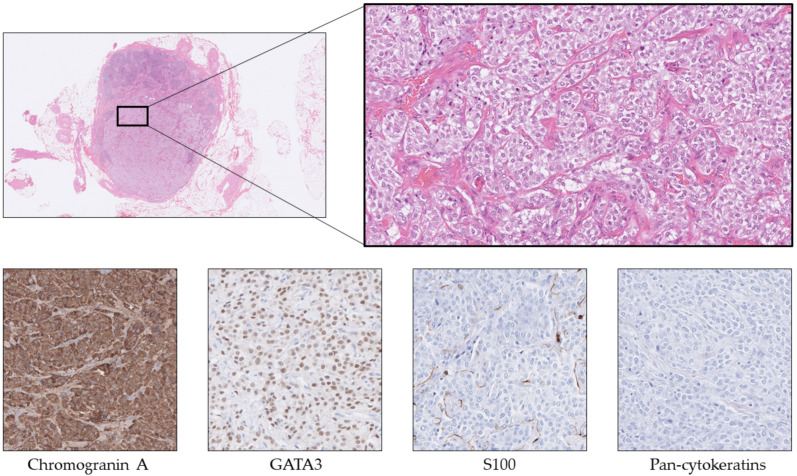
Pheochromocytoma metastatic to a regional lymph node. Note the nested (“zellballen”) appearance of the basophilic tumor cells. The immunohistochemical expression of neuroendocrine markers in combination with GATA3 and concurrent keratin negativity strongly argues in favor of a pheochromocytoma or paraganglioma. Sustentacular cells may be present also in metastatic lesions, as highlighted here by an S100 stain.

**Figure 6 cancers-14-02210-f006:**
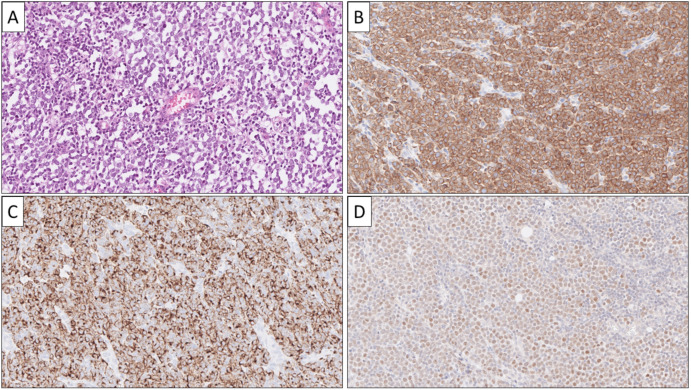
Metastatic Merkel cell carcinoma (MCC). Note the small blue round cell appearance on routine histology (**A**), the diffuse synaptophysin immunoreactivity (**B**), the characteristic, dot-like CK20 stain (**C**), and positivity for MCV-polyoma virus antigen (**D**); the latter may be seen in large subsets of cases.

**Figure 7 cancers-14-02210-f007:**
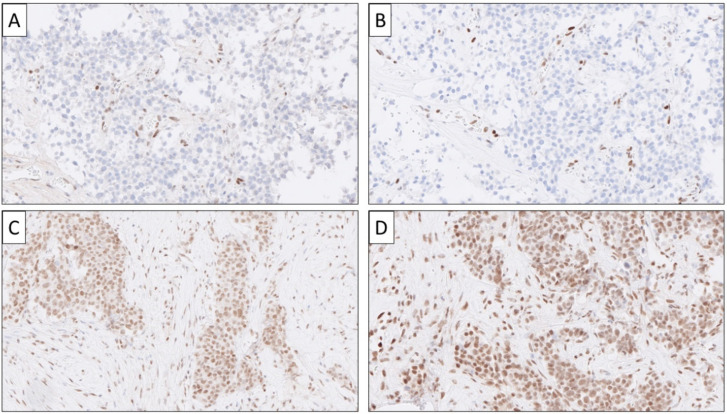
Immunohistochemical expression patterns in pancreatic neuroendocrine carcinomas (Pan-NECs). Recurrent genomic alterations include *TP53* and *RB1* gene mutations, which can be indicated by either complete loss of P53 immunoreactivity (**A**) or alternatively strong and diffuse P53 expression (not shown) compared with the mixed staining pattern noted in *TP53* wild-type cases. Similarly, loss of the Rb protein may reflect an underlying RB1 gene aberration (**B**). Note the retained internal control of the stromal compartment in both A and B. As Pan-NECs are largely driven by *TP53* and *RB1* alterations, they do not harbor *ATRX* or *DAXX* gene mutations as their Pan-NET G3 counterparts. In this Pan-NEC, a nuclear expression for ATRX (**C**) and DAXX (**D**) is diffusely positive, reflecting wild-type genes.

**Figure 8 cancers-14-02210-f008:**
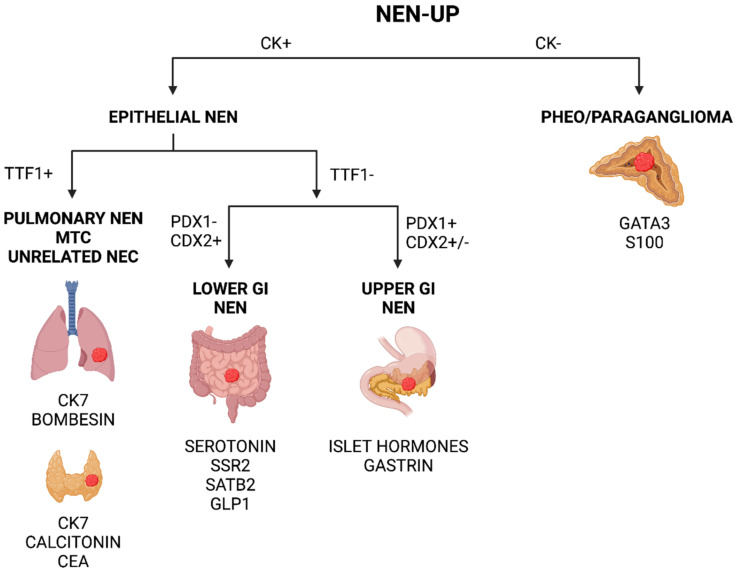
Simplified and generalized scheme to help in the identification of neuroendocrine neoplasia of unknown primary (NEN-UP) using immunohistochemical markers in clinical routine. The algorithm could be applied to tumors of unknown primary that show a clear-cut neuroendocrine differentiation assessed by morphology and immunohistochemistry. Please note that this is a crude scheme that does not categorically identify all NEN-UPs, as, for example, TTF1 may be upregulated in various high-grade lesions, and PDX1 and CDX2 may both be negative in subsets of NENs arising in the upper GI tract. CK—cytokeratins, PHEO—pheochromocytoma, MTC—medullary thyroid carcinoma, GI—gastrointestinal. Created using BioRender.com.

**Table 1 cancers-14-02210-t001:** Histology associated with various neuroendocrine neoplasms.

Primary Site	Common Growth Patterns *	Associated Histological Features
Lung	Various	Rarely with stromal ossification
Thyroid-medullary carcinoma	Various	Amyloid deposits, amphophilic cytoplasm
Thymus	Nested, trabecular, cord-like	-
Stomach	Pseudo-glandular, trabecular, nested	-
Duodenum-somatostatinoma	Pseudo-glandular, nested	Psammoma bodies
Duodenum, Pancreas-gastrinoma	Trabecular, pseudo-glandular	-
Pancreas-insulinoma	Trabecular, nested, solid	Hyalinized stroma
Pancreas-glucagonoma	Nested, cord-like	-
Pancreas-somatostatinoma	Nested, cord-like	Psammoma bodies
Pancreas-VIPoma	Nested, cord-like	-
Pancreas-non-producing	Nested, cord-like	-
Small intestine	Nested, organoid	Peripheral cytoplasmic granularity
Appendix-enterochromaffin	Nested, cord-like	-
Appendix-L-cell	Trabecular, pseudo-glandular	-
Appendix-tubular	Tubular	-
Adrenal-pheochromocytoma	Nested, “zell-ballen”	Hyaline globules, basophilic cytoplasm
Paraganglia-paraganglioma	Nested, “zell-ballen”	Basophilic cytoplasm
Merkel cell carcinoma	Variable	Small round blue cell tumor
Prostate	Small cell phenotype	-
Colon	Nested, trabecular, nested	-
Rectum	Nested, trabecular, cord-like	-
Anal canal	Small cell phenotype	-

* These growth patterns may be observed in both primary and metastatic tumors.

**Table 2 cancers-14-02210-t002:** Immunohistochemical patterns of recognition in neuroendocrine neoplasms.

Primary Site	CGA *	SYP *	ISL1	INSM1	SECG	Other Markers of Importance
Lung	+	+	+	+	+	TTF1, bombesin/GRP
Thyroid-medullary carcinoma	+	+	+	+	N/A	TTF1, PAX8, Calcitonin, CEA
Thymus	+	+	N/A	N/A	N/A	-
Stomach	+	+	N/A	N/A	N/A	PDX1
Duodenum-somatostatinoma	+	+	+	N/A	N/A	PDX1, somatostatin
Duodenum, Pancreas-gastrinoma	+	+	+	N/A	N/A	Gastrin
Pancreas-insulinoma	+	+	+	+	+	Insulin
Pancreas-glucagonoma	+	+	+	+	+	Glucagon
Pancreas-somatostatinoma	+	+	+	+	+	Somatostatin
Pancreas-VIPoma	+	+	+	+	+	VIP
Pancreas-non-producing	+	+	+	+	+	Pancreatic polypeptide
Small intestine	+	+	−	+	+	Serotonin, CDX2
Appendix-enterochromaffin	+	+	+	+	+	Serotonin
Appendix-L-cell	−	+	N/A	N/A	N/A	GLP1, PP, CEA
Appendix-tubular	+	+	N/A	N/A	N/A	Serotonin
Adrenal-pheochromocytoma	+	+	+	+	−	GATA3, S100
Paraganglia-paraganglioma	+	+	+	+	−	GATA3, S100
Merkel cell carcinoma	+	+	+	N/A	N/A	Dot-like CK20, MCV-polyoma
Prostate	+	+	N/A	N/A	N/A	NKX3.1
Colon	+	+	+	+	+	CDX2, SATB2
Rectum	−	+	+	+	+	GLP1, SATB2
Anal canal	+	+	N/A	N/A	N/A	P16

* CGA and SYP immunoreactivity may, in some cases, be lost in neuroendocrine carcinoma; + (positive); − (negative); N/A (information is not available, or only single reports with few cases exist); CGA (Chromogranin A), SYP (Synaptophysin), ISL1 (ISL LIM Homeobox 1); INSM1 (INSM Transcriptional Repressor 1), SECG (Secretagogin). Note that the staining patterns above may be observed in both primary and metastatic lesions.

## Data Availability

Not applicable.

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
