# Peer review of "Metastatic Neuroendocrine Neoplasms of Unknown Primary: Clues from Pathology Workup"

_cancers, 2022, doi:10.3390/cancers14092210_

Round 1

Reviewer 1 Report

General Comments:

The manuscript aims to provide a pathologist overview about the metastatic neuroendocrine neoplasms (NENs) with unknown primary.

The object of the study is very interesting; however, the manuscript has several limitations, in almost all sections.

Please see specific comments.

Specific comments:

Title: ok.

Keywords: ok

Abstract:

  • Please modify the first sentence, it seems quite redundant with the simple summary.
  • Please describe in a more structured manner the main limitations in identifying the primitive tumor.
  • Please enhance in a stronger manner the aim of the review.

Simple summary: Please describe in a more consistent manner the poor aggressiveness of NETs. This aspect seems to be lacking.

Introduction:

  • Please introduce the chance to discover metastatic NENs as incidental tumor, according to poor aggressiveness.
  • What about the sensitivity and specificity of CT and MRI in identify primitive tumor in metastatic NENs? Please discuss.
  • Please rewrite the aim of the study, it seems too general.

Sections 1.1-1.5:

  • Please consider introducing the role of CT, PET and MRI in the diagnostic work-up of NENs before the core needle biopsy, in the diagnostic work-up conventional imaging seems to be the first step before the core needle biopsy.
  • Please add some epidemiological data about the most common NENs.

Discussion and Conclusions:

  • Please avoid mentioning clinical workup in the conclusions, in the main text clinical data and diagnostic workup seems to be lacking.
  • What about “over-interpret”? Please clarify.

References: ok

Tables:

  • Please add some Tables to describe the main findings for each NENs primitive.

Figures:

  • Overall ok, however you might modify the captions in a more synthetic manner.

Linguistic and typewriting: ok.

Reviewer 2 Report

In the the manuscript "Metastatic neuroendocrine neoplasms of unknown primary: Clues from pathology work-up" by C. Christofer Juhlin et al. the authors attempted to summarize new relevant information about the work-up of metastatic neuroendocrine neoplasms of unknown primary in Pathology. The theme is interesting and relevant, and the paper is well-written. However, there are aspects that merit the attention of authors and that need to be improved. The topics of important concern are the following:

  1. While the overall organization of the paper is good, there are some parts that seem to be misplaced. For example, the authors discuss the NENs of the upper GI tract, then jump to lung NENs and paragangliomas; and afterwards comeback to lower GI tract NENs. It would be advisable to place the lower GI tract section after the upper GI tract section.
  2. The authors should make more clear in the text that most of these markers have an important degree of non-specificity in the context of NENs and, thus, their interpretation should always be cautious (for example, CDX2 and TTF1).
  3. In line with the previous comment, Figure 8 can be misleading, since TTF1 can be positive in NENs from different locations. As it stands, it looks like that all GI tract NENs are TTF1 negative, which is not the case. This figure should be reformulated or eliminated.
  4. The format of the legends of the figures should be uniform and allow the reader to easily identify the markers being used in the panels. That is not currently the case. For example, in Figure 1, the description “trabecular growth pattern on routine staining (A), and was positive for Chromogranin A, PDX1 and somatostatin (B-D)” might be confusing for some readers.
  5. The authors should discuss the new developments of molecular pathology in the identification of the primary site of NEN-UPs. Does Next Generation Sequencing (NGS) might play an helpful role?

Therefore, it will be mandatory to perform additional changes in the present manuscript if it is considered relevant for publication in Cancers.

Round 2

Reviewer 1 Report

the manuscript has been properly improved

Reviewer 2 Report

In the revised version of manuscript "Metastatic neuroendocrine neoplasms of unknown primary: Clues from pathology work-up" by C. Christofer Juhlin et al. the authors attempted to summarize new relevant information about the work-up of metastatic neuroendocrine neoplasms of unknown primary in Pathology. The theme is interesting and relevant, and the paper is well-written. The authors addressed all the points raised in the previous version of the paper and, thus, I strongly recommend its publication in Cancers in the present format.